# Comparing the Impact of Non-Gamified and Gamified Virtual Reality in Digital Twin Virtual Museum Environments: A Case Study of Wieng Yong House Museum, Thailand

Suepphong Chernbumroong [1], Pakinee Ariya [1], Suratchanee Yolthasart [2], Natchaya Wongwan [1], Kannikar Intawong [3] and Kitti Puritat [4,*]

1   College of Arts, Media and Technology, Chiang Mai University, Chiang Mai 50200, Thailand; suepphong.c@cmu.ac.th (S.C.); pakinee.a@cmu.ac.th (P.A.); natchaya_won@cmu.ac.th (N.W.)
2   School of Tourism, Chiang Rai Rajabhat University, Chiang Rai 57100, Thailand; suratchanee.yol@crru.ac.th
3   Faculty of Public Health, Chiang Mai University, Chiang Mai 50200, Thailand; kannikar.i@cmu.ac.th
4   Faculty of Humanities, Chiang Mai University, Chiang Mai 50200, Thailand
*   Correspondence: kitti.p@cmu.ac.th

**Abstract:** Virtual reality (VR) is increasingly employed in various domains, notably enhancing learning and experiences in cultural heritage (CH). This study examines the effects of gamified and non-gamified VR experiences within virtual museum environments, highlighting the concept of a digital twin and its focus on cultural heritage. It explores how these VR modalities affect visitor motivation, engagement, and learning outcomes. For this purpose, two versions were developed: a gamified virtual reality version incorporating interactive gaming elements like achievements, profiles, leaderboards, and quizzes and a non-gamified virtual reality version devoid of these elements. This study, using an experimental design with 76 participants (38 in each group for the gamified and non-gamified experiences), leverages the Wieng Yong House Museum's digital twin and its fabric collection to assess the educational and experiential quality of virtual museum visits. The findings indicate that while gamification significantly boosts the reward dimension of visitor engagement, its influence is most pronounced in the effort dimension of motivation; however, its impact on learning outcomes is less marked. These insights are instrumental for integrating VR and gamification into museum environments.

**Keywords:** virtual museum; digital twin; culture heritage; gamification; virtual reality

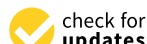



## 1. Introduction

In recent years, virtual reality (VR) technology has not only gained substantial recognition for its potential to enhance visitor experiences in museums and cultural heritage destinations [1–4] but has also led cultural heritage organizations to specifically invest in VR as a means with which to overcome the physical environmental limitations of cultural heritage configuration/exhibition, thereby enriching visitor learning experience by virtualizing and augmenting exhibits in innovative ways [5,6]. A notable example is the reopening of the Domus Aurea construction site in 2017, where an innovative exhibition incorporated site-specific VR projects and video mapping [7]. Likewise, in 2019, the Louvre introduced the first VR experience for the Mona Lisa, titled 'Mona Lisa: Beyond the Glass', which was also accessible on smartphones. This VR experience provided a detailed insight into Leonardo da Vinci's painting techniques, bringing his masterpiece to life. The immersive journey began in the modern Louvre's Salle des États, allowing visitors to face the iconic Mona Lisa before transporting them back in time [8]. Another noteworthy instance is the Museo Nazionale Della Scienza e Della Tecnica in Milan, which, since February 2022, has incorporated a virtual reality zone within the rooms of the old museum cinema [9]. Consequently, VR enhances the value of a museum exhibition using unusual narratives such

as interactive storytelling, first-person perspectives, immersive environments, non-linear time exploration, and emotional and sensory experiences that combine experiential learning with innovative entertainment recreation and other typologies of experiences [10,11]. These types of narratives enhance the value of a museum by offering unique, engaging experiences that are not possible through traditional exhibits. They can make learning more interactive and enjoyable, increase visitors' engagement and retention of information, and attract a wider audience, including those who might not typically be interested in museums. Additionally, by offering personalized and deeply immersive experiences, VR can help museums stand out in a crowded entertainment and education landscape, encouraging repeat visits and word-of-mouth promotion [12]. Simultaneously, the positive consequences of employing VR in these contexts include enriched learning outcomes, heightened satisfaction, and positive emotional responses, such as pleasure and enjoyment [1]. Moreover, as VR technology becomes more affordable and accessible for personal use, it creates new opportunities for active participation in cultural heritage through virtual experiences. These digital engagements offer immersive interactions, making cultural heritage more accessible and educational. They allow for the exploration of remote or inaccessible sites, provide interactive learning environments, and support the preservation and virtual reconstruction of endangered or lost cultural artifacts. Digital twins, a concept that has garnered significant attention in recent years, refer to a computational model capable of conducting analyses and predictions regarding the behavior of physical assets under varying conditions [13], emphasizing the significance of dynamic data collection, such as forces, loads, temperature variations, and rate-dependent phenomena, from the physical asset to its digital counterpart [14]. Expanding on the immersive capabilities of VR, the adoption of digital twins for cultural heritage goods offers a transformative approach. This technology creates detailed, dynamic digital representations of cultural artifacts and sites, enabling real-time interaction, predictive maintenance, and a deeper understanding of historical contexts. By incorporating physical properties and dynamic changes, digital twins can provide a more nuanced and educational exploration of cultural heritage, enhancing conservation efforts and the visitor learning experience [15]. Additionally, the integration of gamification elements within VR environments has emerged as a noteworthy trend, offering a unique approach to education and engagement that bridges the virtual and physical realms. For instance, in 'A Night in the Forum', a PlayStation® VR game that transports players to ancient Rome during the Augustan era, participants are immersed in the Forum of Augustus through environmental storytelling and learning-by-doing methods. Assuming the role of a guardian, players complete various tasks and interact with digital replicas of real cultural artifacts. This game demonstrates that the combination of VR and gamification can enrich engagement with cultural heritage by immersing participants in authentic historical scenarios and challenges. Similarly, the VR puzzle game known as μVR Forum offers users the challenge of recontextualization within the architectural and archaeological settings of the Forum of Augustus. By integrating elements of real-world navigation and multi-scale 6DOF gaming, players are tasked to find and accurately position misplaced items throughout the forum. This innovative approach highlights the potential of VR and gamification in facilitating educational experiences, providing an immersive and engaging way to learn about historical architecture and artifacts [16]. Considerable research has demonstrated the benefits of gamification: for instance, a meta-analysis examining the impact of gamification on behavioral change in education revealed moderately positive effect sizes of gamification on learning outcomes [17]. However, it has also given rise to controversies and criticisms, with concerns raised about potential performance decline and a gradual loss of motivation [18]. In light of these considerations, our research aims to delve into a comparative analysis of two key approaches in digital twin virtual museum environments, non-gamified and gamified VR, by assessing their impact on motivation, user engagement, and learning outcomes.

The research is structured as follows: Sections 1 and 2, 'Introduction' and 'Literature Review', respectively, provide a background and analyze previous work relevant to the

topic. The objectives of the research are specified in Section 3, 'Purpose of the Study'. The 'Development Process' and 'Research Methodology' sections, encompassing Sections 4–9, explain the design and development of the virtual reality experience for the FabricVR project, along with the methodologies employed. The 'Data Analysis and Results' section, presented in Section 10, summarizes the data, provides an in-depth discussion, and presents important findings. Finally, the 'Conclusions' section includes concluding remarks, potential directions for future research, and an acknowledgment of the study's limitations.

## 2. Literature Review

### 2.1. Virtual Reality

In recent years, there has been a growing interest among museums in incorporating virtual reality (VR) [6,19,20]. Virtual reality (VR) serves as a computer-simulated 3D environment, generating digital representations of multisensory virtual worlds to augment museum content. Within this virtual environment, museum visitors actively engage, encountering interactive perceptions and diverse illusory elements of socialization [6,21–23]. When using VR, visitors typically utilize wearable devices that block out reality, transporting them into a virtual 3D world [24,25]. This technology enables users to experience distant worlds, ancient places, or rare and immobile exhibits, providing a sense of being in an artificial virtual environment [26,27]. Researchers have highlighted the educational potential of VR, emphasizing its use in promoting learning and motivating students [28,29]. Moreover, scholars have demonstrated that well-developed VR content not only motivates students but also allows for the in-depth exploration of a topic. For example, the authors of [30] selected university students as a demographic for exploring VR's potential in education due to their cognitive and technical readiness, the relevance of VR applications to their academic and professional pursuits, and the natural fit between advanced educational objectives and the capabilities of VR technology. To further illustrate the practical application of this technology in higher education settings, by designing simulations with game-like features, university students are likely to develop positive learning behaviors and feel more inclined to participate actively in the learning process. The four critical elements of experiencing VR, virtual space, immersion, sensory feedback, and interactivity, contribute to creating an environment where students can feel part of the virtual world. This immersion helps maintain students' interest and motivation throughout the learning process. Additionally, by providing a platform where students can experiment and take risks without real-world consequences, VR encourages creative thinking and problem-solving skills. This environment supports in-depth exploration, allowing students to test various hypotheses and solutions in their quest for understanding. Additionally, as a promotional tool, VR can motivate individuals to physically visit traditional museums [23]. Recognizing the diverse needs of museum visitors, who seek entertainment, relaxation, leisure, and spiritual as well as social experiences, among other things [31], VR has been acknowledged as a compelling means with which to enable museum visitors to experience places or objects that cannot be physically exhibited, reconstructed, or reenacted due to budgetary constraints, limited space, or staffing issues [32]. To highlight some studies that applied VR in museums, Lee et al. [23] conducted an assessment of the role of VR in influencing intentions to visit a museum. Their findings indicated that VR users reported increased intentions to visit. This is attributed to VR's ability to offer both educational and entertaining content, along with heightened levels of immersion. These factors collectively contribute to an enhanced overall museum experience and influence the intention to visit a museum. Moreover, Jung et al. [33] investigated how virtual reality (VR) and augmented reality (AR) contribute to encouraging a desire to revisit the Geevor Tin Mine in the UK. Through VR, the researchers examined an immersive experience involving a non-functional lift descending into a mine to illustrate the commencement of miners' work. Notably, although the actual lift is no longer operational, visitors can still encounter it through VR; they also demonstrated that VR could create a positive visiting experience, subsequently increasing the intention to revisit a museum. Furthermore, Tennent et al. [34] created a VR recreation of the world's

first photographic exhibition and exhibited it globally across multiple museums. Their study revealed that enhancing VR content with additional sensory elements (e.g., heat, smell, touch, and movement) contributes to the enhancement of museum experiences.

*2.2. Gamification*

In recent years, gamification has attracted considerable attention as an innovative approach. It involves the integration of game elements like rewards, missions, and rankings in non-gaming environments such as education, management, healthcare, and tourism, usually to encourage students' engagement with a product or service [35]. Importantly, the concept of gamification does not necessarily involve developing actual games; instead, it involves the utilization of playfulness and playful strategies to create engaging experiences with the purpose of achieving a specific objective [36]. Within gamification, a diverse range of game design patterns are employed. Werbach and Hunter [37] identified 15 common game design elements, with badges, rewards, leaderboards, feedback, missions, and progress being commonly mentioned. By incorporating these elements, gamification enriches services and systems, generating experiences that are similar to those found in games. These game-like experiences play a crucial role in encouraging and motivating users to engage in purposeful actions and behaviors, fostering positive attitudes towards services and active participation in learning activities [38–40]. Importantly, the implementation of gamified practices enables individuals to derive enjoyment, experience flow, exercise autonomy, attain mastery, and achieve a sense of accomplishment through diverse game design elements, like missions and quizzes [41]. Notably, Aparicio et al. [42] proposed a gamification framework designed to enhance participation and motivation in various tasks. The researchers recommended strategies for motivating individuals through game mechanics, such as points, levels, and leaderboards, emphasizing autonomy, competence, and relatedness.

Gamification has emerged as a transformative and versatile tool, finding applications in diverse fields, such as education, training, health, self-management, innovation, employee engagement, and heritage [43]. Expanding upon its widespread application, recent studies have further described gamification's impact on educational outcomes. Öztürk and Korkmaz suggested that gamification significantly enhances students' attitudes towards social studies courses compared to traditional teaching methods. Additionally, their study found that gamification notably improves students' cooperative learning skills and academic achievement in social studies [44]. Lister indicated that gamification elements like points, badges, and leaderboards, when effectively implemented, can significantly motivate students and support student achievement in post-secondary environments. The study reported increased class attendance and participation, positively correlating with improved student performance due to gamification strategies [45]. Further supporting these findings, Papp and Theresa concluded that gamification effectively increases student motivation, engagement, and learning outcomes across the primary and college levels [46]. Moreover, Dicheva et al. emphasized the need for broader adoption and investigation into gamification's feasibility and efficacy across various educational domains, suggesting that while the application of gamification in education shows promise, effective implementation is crucial for its success [47]. Legaki et al. observed that challenge-based gamification improved student learning compared to traditional methods, suggesting enhanced engagement and better outcomes in statistics education. The gamified approach not only motivated students but also led to superior academic performance compared to traditional methods or reading exercises alone [48].

In heritage fields, its applications range from marketing tourist destinations [49] to safeguarding intangible and digital heritage assets [50], in addition to engaging in participatory methods to address challenging aspects of heritage [51]. Transitioning to a more specific instance of gamification's impact, O'Connor et al. discovered that digital games, especially virtual reality games, are effective in engaging individuals in learning about cultural heritage. For instance, I-Ulysses, a virtual reality game based on James

Joyce's Ulysses, was positively received for its gamified mechanics and educational value, with feedback from focus groups showing that I-Ulysses provides an informative and engaging guide to Ulysses, appealing to a wide audience [52]. Further supporting the potential of gamification in heritage education, Xu et al. [41] proposed gamification as an innovative approach to mobile-based learning within the tourism domain. They emphasized the significance of incorporating gamification design elements as tools to amplify the technology's impact on motivating and influencing visitor behavior. Furthermore, Sigala [53] demonstrated the beneficial impact of gamification on motivational behavior, such as increased participation and involvement, as well as on psychological outcomes. The study compared the behavioral perceptions of gamified app services between users and non-users on the TripAdvisor platform within a virtual community. Likewise, the authors of [54] highlight that the implementation of gamification has the potential to enhance engagement, effectively meeting the increasing expectations of tourists. Additionally, the authors of [55] emphasized that gamification enhances learning experiences in museums by guiding visitors toward specific learning objectives during their visits. However, it is essential to recognize that gamification, despite its positive effects, is highly dependent on the context of its implementation and the individual qualities of a user. Improperly executed gamification can lead to unfavorable results [56]. Furthermore, the study also suggests that the impacts of gamification may be temporary, possibly due to the effects of novelty and curiosity. An overview of related studies investigating the integration of VR and gamification is described in Table 1.

**Table 1.** An overview of related studies on VR and gamification.

| Authors | Year | VR Technology/Functionality | VR Game Types | Results |
|---|---|---|---|---|
| Tredinnick and Richens | 2015 [57] | Dome projections, caves, and holographs | Serious game, interactive storytelling, and co-op multiplayer | The use of fulldome projection spaces allows for a highly immersive and personalized experience. As visitors are enveloped in a virtual environment, the content can be tailored to their movements and interactions, making the learning experience more direct and personal. |
| Heryadi et al. | 2016 [58] | Cardboard–mobile usage, other | Serious game and beat 'em up game | Optimizing VR game experiences based on playing frequency and personality traits can enhance user engagement and satisfaction. |
| Li and Zhou | 2016 [59] | HMDs and hand or body tracking | Serious game and co-op multiplayer | The exhibit represents a significant advancement in utilizing VR technology for science popularization in museums, offering an engaging and informative experience for visitors. |
| Lacono et al. | 2018 [60] | HMDs | Serious games, as well as escape, puzzle, and quest games | By combining immersion, interactivity, and the inherently enjoyable nature of arcade games, this study achieves significant learning effects, including raising awareness, facilitating learning gains, and enhancing enjoyment in addition to engagement among players. |
| Vu et al. | 2018 [61] | HMDs | Serious game, interactive storytelling, and escape, puzzle, and quest games | The study suggests that serious VR games have the potential to be effective tools for preserving cultural heritage, educating individuals about historical events and communities, and offering engaging learning experiences that can enhance understanding and appreciation of the past. |
| Martyastiadi | 2020 [62] | HMDs | Serious game, interactive storytelling, and escape, puzzle, and quest games | Incorporating spiritual elements from the Borobudur temple into interactive digital art, offering users a unique and immersive experience that combines aesthetics, spirituality, and technology in the virtual reality realm. |

**Table 1.** *Cont.*

| Authors | Year | VR Technology/Functionality | VR Game Types | Results |
|---------|------|------------------------------|---------------|---------|
| Fu et al. | 2020 [63] | HMDs, multimodal interfaces, and eye tracking | Serious games, as well as escape, puzzle, and quest games | The study suggests that the integration of BCI and VR technologies can significantly enhance cultural experiences in gaming environments, offering new possibilities for immersive and interactive applications in the fields of cultural heritage and entertainment. |
| Rahimi et al. | 2020 [64] | HMDs | Serious game and interactive storytelling | Integrating VR technology into museum experiences not only significantly enhances enjoyment and engagement among visitors but also leads to reported increases in knowledge and learning after engaging with the VR-enhanced museum environment, particularly in understanding specific events and themes related to North American urban history. |
| Liu et al. | 2021 [65] | HMDs | Serious games, as well as escape, puzzle, and quest games | Players found RelicVR more interesting than conventional museum visits, as it allowed for closer interaction with artifacts and provided a deep impression of relics and their information. |
| Zhang et al. | 2021 [66] | HMDs and hand or body tracking | Serious game and interactive storytelling | Integrating traditional art techniques into modern VR gaming can offer unique and enriching experiences. This innovative approach not only makes cultural art forms more accessible to the general public but also opens up new possibilities for interactive design in VR applications. |
| Egea-Vivancos and Arias-Ferrer | 2021 [67] | HMDs | Serious game and interactive storytelling | Incorporating civic education, historical relevance, engagement, applicability, and multimodality (CREAM model) in educational VR video game design effectively enhances educational outcomes. |
| Baradaran Rahimi et al. | 2022 [68] | HMDs | Serious games, as well as escape, puzzle, and quest games | VR and hybrid spaces have significant potential to revolutionize museum experiences, making them more engaging, educational, and accessible beyond the physical constraints of traditional museum walls. |

## 3. Purpose of Study and Research Questions

The primary purpose of this study is to critically examine and compare the effects of non-gamified and gamified virtual reality (VR) experiences on visitors within digital twins of museum environments, particularly in the context of cultural heritage. This comparison aims to explore three main aspects: motivation, user engagement, and the learning outcomes related to the cultural heritage of ancient fabric, specifically in the context of Yok Dok weaving crafts. A museum provides information on the historical context, craftsmanship, and cultural value of these crafts. By investigating these elements, this study seeks to provide insights into how different VR approaches influence the educational and experiential qualities of museum visits in a digital setting.

**(RQ1).** *Are there any differences in how gamified and non-gamified VR experiences in digital twin museum environments affect visitor motivation?*

**(RQ2).** *Are there any differences in visitor engagement between gamified and non-gamified VR experiences in digital twin museum environments?*

**(RQ3).** *Are there any differences in the impact of gamified versus non-gamified VR experiences on learning outcomes related to cultural heritage in digital twin museum environments?*

In this study, we aim to explore the differential impacts of gamified and non-gamified virtual reality (VR) experiences in digital twin museum environments through a series of focused research questions. The first question, RQ1, investigates whether notable differences exist in how gamified versus non-gamified VR experiences affect visitor motivation. This inquiry seeks to understand if the integration of game elements in VR can significantly enhance the motivational aspects of museum visits. Subsequently, RQ2 examines the differences in visitor engagement elicited by gamified and non-gamified VR. Here, the objective is to determine if gamification influences the level and nature of engagement in a museum's digital twin environment. Finally, RQ3 delves into the impact of these two VR modalities on learning outcomes related to cultural heritage. This question is pivotal in assessing whether gamified VR experiences offer distinct educational benefits compared to non-gamified VR, particularly in terms of effectively conveying cultural heritage in a virtual museum setting.

## 4. Virtual Reality Design and Implementation

In the realm of digital preservation and interactive museum experiences, the Wieng Yong House Museum has been reconstructed conceptually as a digital twin [58,59], a key component of the FabricVR project [57]. This project focuses on digitizing the museum's ancient fabric collection and visualizing cultural resources on a digital platform, thereby significantly enhancing the virtual museum experience for visitors. Central to the museum's mission is the preservation and celebration of weaving crafts, with particular emphasis on 'Yok Dok', a renowned woven fabric from Thailand, as illustrated in Figure 1. The development of virtual reality technology within this initiative introduces an innovative dimension to the museum's unique and valuable textile collection. As a custodian of traditional crafts, the Wieng Yong House Museum plays an integral role in preserving textile artisanship. It has established community-based handicraft centers in Lamphun, thereby maintaining the continuity and vibrancy of local textile traditions.

**Figure 1.** FabricVR Project: integrated workflow and project goals.

## 5. Selection of Hardware and Software

In the development of the virtual reality experience for the FabricVR project, a combination of advanced software and hardware technologies was meticulously selected to achieve a high level of realism and interactivity. The Unity Game Engine, specifically the 2021 version, served as the cornerstone for VR development, providing a robust platform for integrating complex functionalities and interactive features essential for a virtual museum experience. Concurrently, Blender version 3.6 was employed for the creation of intricate 3D models, while Adobe Photoshop version 22.5.2 was used for designing textures. These textures were then accurately applied to the models through UV mapping in Blender.

Additionally, RealityCapture version 1.3 played a pivotal role in photogrammetry, enabling the transformation of photographs into detailed 3D models, a process crucial for digitally representing the museum's fabric collection with high fidelity.

On the hardware front, the Meta Quest 2 headset (manufactured by Meta Platforms, Inc., Menlo Park, CA, USA) was selected for its capability to offer an accessible and deeply immersive VR experience. The project's computing demands were met by a high-end computer equipped with an Intel i7 2.6 GHz 16-core CPU and 32 GB of RAM (manufactured by Intel, Santa Clara, CA, USA). The use of an NVIDIA RTX 3080 graphics card (manufactured by NVIDIA, Santa Clara, CA, USA) was crucial for visualizing graphic models and rendering high-fidelity visuals, which are critical to achieving a realistic and immersive virtual experience of the museum.

## 6. Reconstruction and Digitalization of Heritage Objects in Museums

The FabricVR project's virtual environment represents a seamless integration of photogrammetry with reconstructed 3D models, resulting in a highly realistic portrayal of the Wieng Yong House Museum. This digital reconstruction was meticulously carried out using blueprint maps (referenced in Figure 2), along with various heritage objects from the museum, ancient fabrics, detailed photographs, and models provided by the curators of the Wieng Yong House Museum. Their collaboration played a crucial role in the successful realization of the FabricVR project.

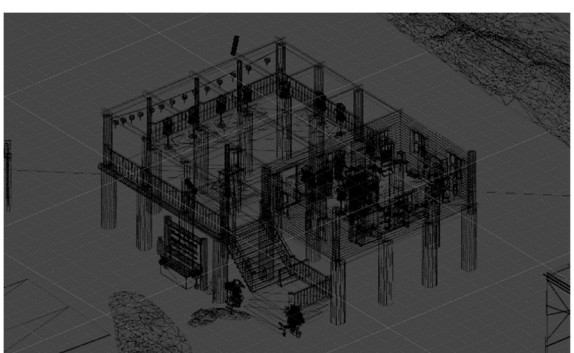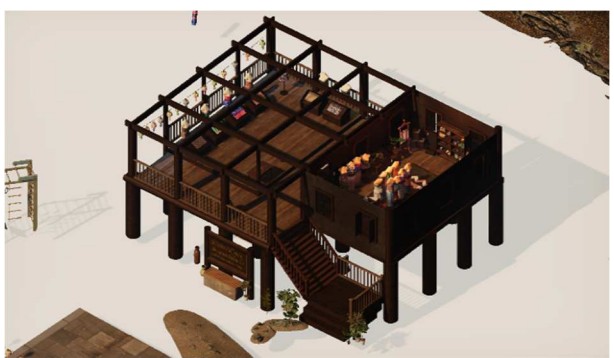

**Figure 2.** The digital blueprint and reconstructed 3D models of the Wieng Yong House Museum.

In alignment with the concept of a digital twin, the digitalization of heritage objects in the Wieng Yong House Museum was systematically categorized into three main types: (1) the building of the museum, (2) ancient fabric, and (3) heritage objects in the museum, as detailed in Table 2. This methodical approach enabled a comprehensive and precise digital reconstruction, ensuring the authentic representation of each element in the virtual environment.

Within the museum, we digitally reconstructed sections of the Wieng Yong House Museum, including floors one and two. This allows users to freely explore the museum using a Meta Quest 2 motion controller(manufactured by Meta Platforms, Inc., Menlo Park, CA, USA). Users can navigate the museum through a locomotion method [56], which is suitably designed for navigating cultural heritage spaces. In the 'walkable' areas of the museum, each digitally reconstructed heritage object is interactive, allowing users to access the full information about each item (as shown in Figure 3). Furthermore, for ancient fabrics, users have the ability to interact in a manner akin to real fabrics. This is achieved through the employment of a physical cloth simulation method [55], enabling users to pull, crush, and fold the fabric, as demonstrated in Figure 4.

**Table 2.** Digitalization of heritage objects in the Wieng Yong House Museum.

| Type of Digitalization | Digitalization Method | List of Heritage Objects | Description |
|---|---|---|---|
| Building of the museum | Blueprint map, sketch, and cooperation with museum curators | Wieng Yong House Museum floors one and two | Digitally recreated using original blueprints and curatorial input, providing an accurate virtual representation of the museum's architecture. |
| Ancient fabric | Fujitsu ScanSnap SV600 image scanner and a Cannon EOS600D kit 18–55 | Yok Dok woven textile (nine pieces) | High-quality scans captured the intricate patterns and vibrant colors of Yok Dok, a traditional Thai brocade. |
| | Fujitsu ScanSnap SV600 image scanner and a Cannon EOS600D kit 18–55 | Raised pattern weaving (six pieces) | Scanned with precision to highlight the unique raised textures and traditional designs of these woven fabrics. |
| | Fujitsu ScanSnap SV600 image scanner and a Cannon EOS600D kit 18–55 | Sarong (eight pieces) | Each sarong's distinctive patterns and cultural relevance were carefully digitized, reflecting regional textile artistry. |
| Heritage objects in the museum | Three-dimensional modeling and iPhone Pro Max 13 | Ancient old pictures (five pieces) | Historical photographs transformed into high-resolution digital formats, preserving their historical essence. |
| | Photogrammetry with RealityCapture software version 1.3 | Ancient Buddha statue (four statues) | Statues meticulously digitized, showcasing intricate details and craftsmanship of Buddhist art. |
| | Photogrammetry with RealityCapture software version 1.3 | Fitting mannequin (10 pieces) | Detailed digital models of mannequins provide insights into historical fashion and garment display. |
| | Photogrammetry with RealityCapture software version 1.3 | Ancient typewriter | The typewriter's mechanical complexity and historical significance were captured in a detailed 3D model. |
| | Three-dimensional modeling and iPhone Pro Max 13 | Other furniture (20 pieces) | A diverse collection of digitized antique furniture highlighting varied styles and eras of craftsmanship. |
| | Photogrammetry with RealityCapture software | Earthenware (10 pieces) | Traditional earthenware pieces scanned to capture their unique designs and cultural importance. |

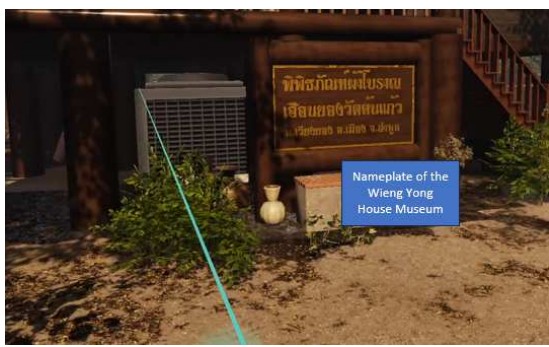 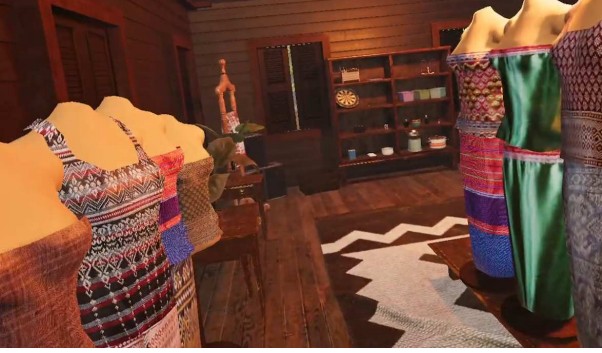

**Figure 3.** Participants navigate the 3D digital museum and access complete information.

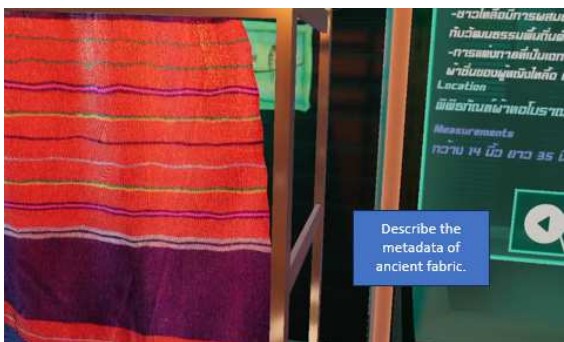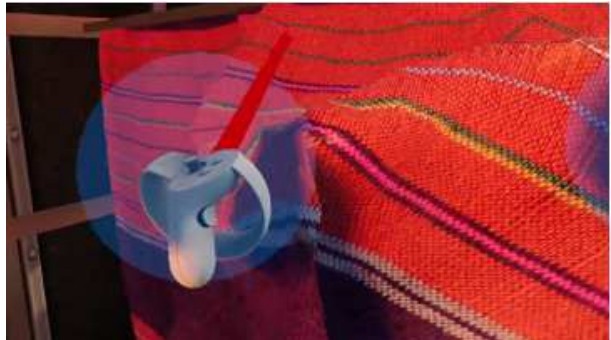

**Figure 4.** Participants interact with a physical simulation of ancient fabrics, enabling them to pull, crush, and fold them.

## 7. Non-Gamified Virtual Reality and Gamified Virtual Reality Version

In our endeavor to assess the impact of gamification on virtual reality experiences within digital twins of museums, we developed two distinct versions of virtual reality applications: non-gamified and gamified versions. The non-gamified virtual reality version emphasizes an informative and immersive experience, focused exclusively on educational content and visual immersion, and is devoid of any gaming elements. In contrast, the gamified virtual reality version is enriched with interactive gaming elements specifically designed to augment visitor engagement and enhance educational outcomes.

In the gamified version, 'Achievements' [60], recognitions or badges awarded for completing tasks or reaching milestones, are employed to motivate users to thoroughly explore the museum and actively engage with its exhibits. 'Profiles' [61], which are personalized user interfaces displaying achievements, preferences, and history, are designed to provide a customized experience and track user progression. 'Leaderboards' [62,63], presenting user scores or achievements in comparison to others, are used to instill a competitive spirit and encourage deeper interaction with the VR content. 'Progression' [61], a feature that unlocks new levels or content as users advance, aims to sustain user interest by continuously introducing new elements. Lastly, 'Quizzes' [64,65], comprising interactive questionnaires or puzzles related to a museum's content, are integrated to enhance the learning and retention of information about a museum's exhibits. A summary of these game elements applied is shown in Table 3, and screenshots of the gamified version in use are depicted in Figures 5 and 6.

**Table 3.** Game elements applied in the gamified virtual reality version of the Wieng Yong House Museum.

| Game Elements | Definition | Objective |
| --- | --- | --- |
| Achievement | Recognitions or badges awarded to users for completing specific tasks or reaching certain milestones [60]. | To motivate users to explore more sections of the museum and engage deeply with the exhibits. |
| Profile | A personalized user interface that displays the user's achievements, preferences, and history [61]. | To provide a personalized experience and track user progress as well as interactions. |
| Leaderboard | A ranking system that displays user scores or achievements compared to other users [62,63]. | To encourage a competitive spirit and incentivize users to engage more with the VR content. |
| Progression | A system that allows users to unlock new levels or content as they advance in the experience [61]. | To maintain user interest and engagement over time by gradually introducing new content. |
| Quiz | Interactive questionnaires or puzzles related to the museum content [64,65]. | To enhance the learning and retention of information about the museum's exhibits. |

These gamified elements are strategically integrated into the virtual reality version to enhance user engagement and support learning about ancient fabrics and their preservation methods. This approach is intended to create an interactive, dynamic learning environment,

potentially amplifying the effectiveness and appeal of cultural heritage education in a digital realm.

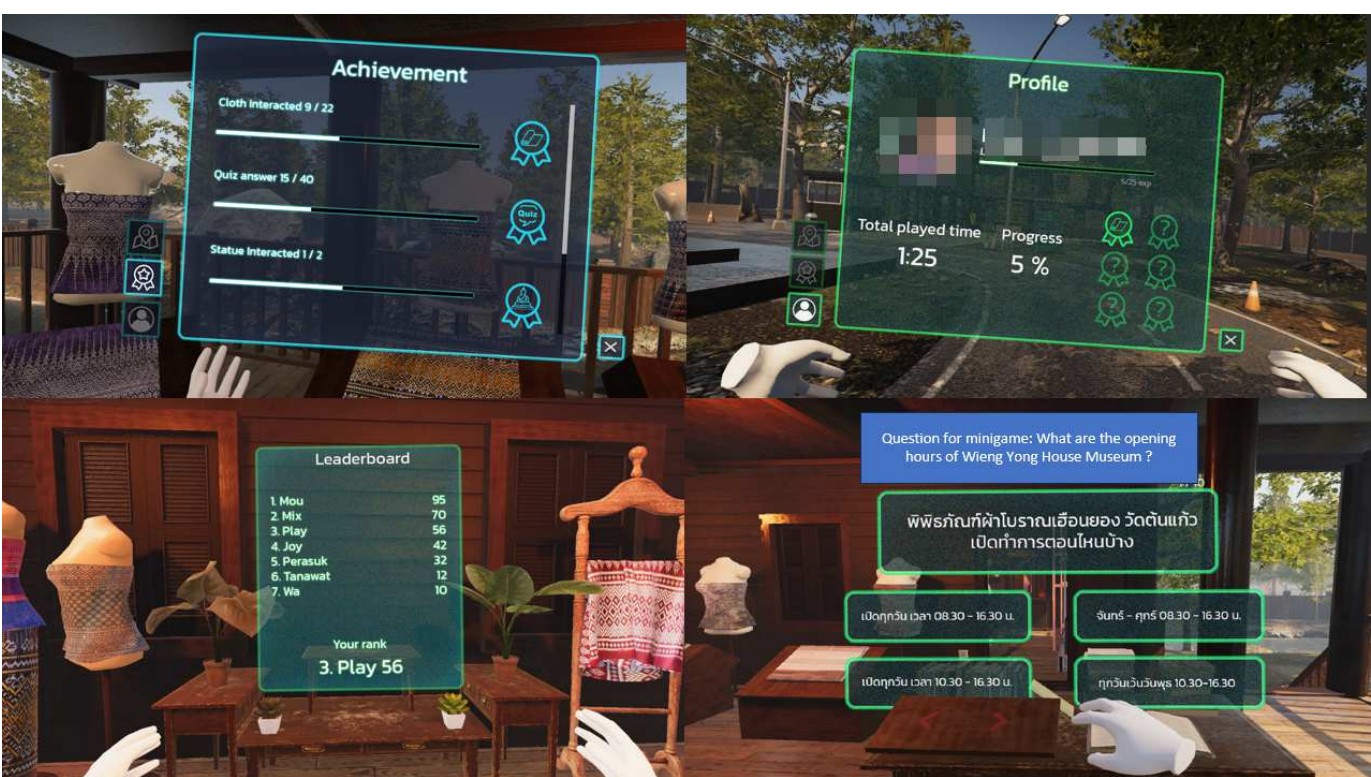

**Figure 5.** Gamified virtual reality incorporating game elements: achievement, profile, leaderboard, and quiz.

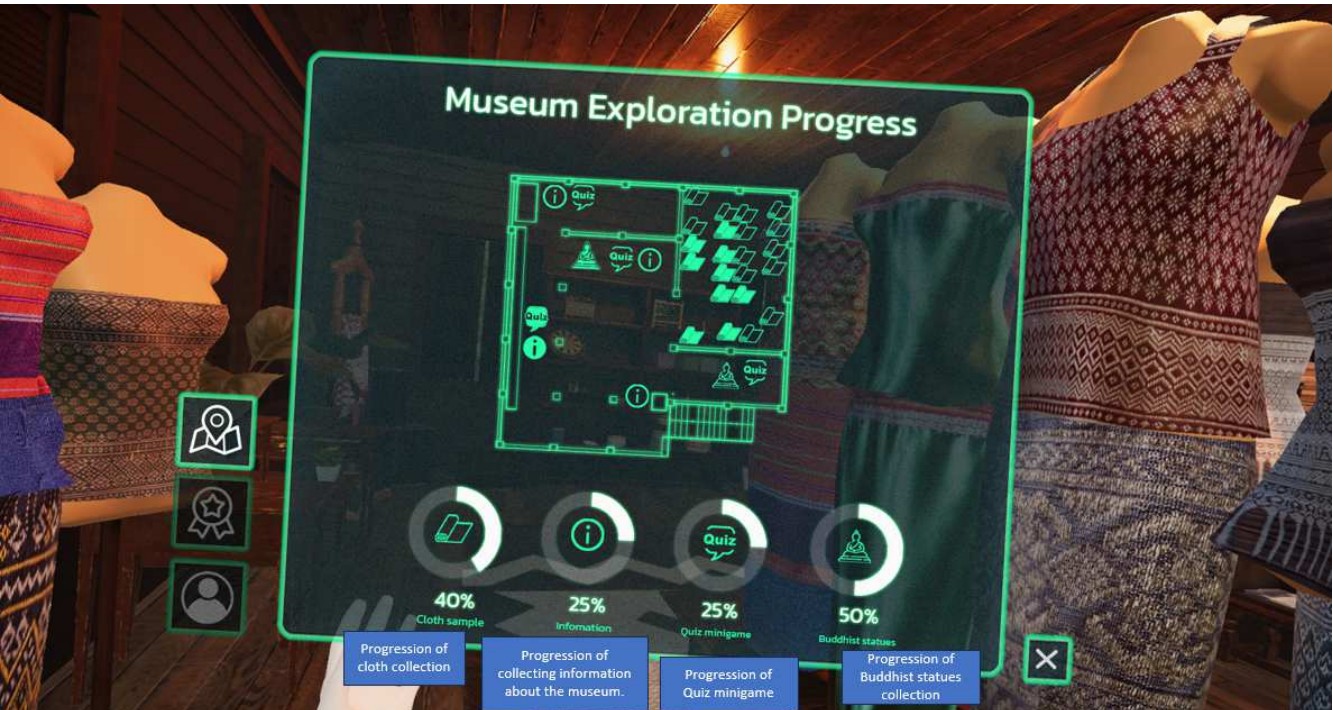

**Figure 6.** Gamified virtual reality applied the progress map in the museum exploration progress.

## 8. Research Methodology

In this study, we adopt an experimental design to assess the impact of gamification in virtual reality (VR) experiences within digital twins of museums. The focus is to compare two distinct VR versions: non-gamified virtual reality and gamified virtual reality. This design allows for a controlled comparison with random assignment to either group, enhancing the validity of the results. The experimental design involves randomly assigning participants to one of the two VR experiences: non-gamified or gamified.

*Participants*

Participants for this study were selected from undergraduate students at the Faculty of Humanities, Chiang Mai University. A total of 76 students were recruited and randomly assigned to one of two groups: the non-gamified VR group or the gamified VR group. This random assignment is critical to ensure the comparability of the groups and the validity of the study's findings. A primary selection criterion was familiarity with virtual reality head-mounted displays (HMDs). To mitigate potential biases in the study, candidates who experienced VR-related issues, such as motion sickness, nausea, or dizziness, were excluded. The participant pool had an average age of 22.8 years, with a standard deviation of 0.34 years, suggesting a relatively uniform age distribution. The composition of the sample was 30 male and 46 female participants, offering a balanced gender representation. Detailed demographic data for each evaluation group are comprehensively presented in Table 4.

**Table 4.** Demographic of participants.

| Categories | Statistics of the Pooled Sample |
|---|---|
| Sample size (N) | 76 |
| Mean age (S.D.) | 22.8 (0.34) |
| Male (%) | 30 (39.47%) |
| Female (%) | 46 (60.52%) |
| Never visited the Wieng Yong House Museum (%) | 74 (97.36%) |
| Visited the Wieng Yong House Museum (%) | 2 (2.63%) |

## 9. Instrument

### 9.1. Non-Gamified and Gamified Virtual Reality for Museums

As outlined in a previous section, we developed two versions of virtual reality (VR) applications tailored for museum experiences. The gamified virtual reality version incorporates interactive gaming elements such as achievements, profiles, leaderboards, and quizzes. These elements are intentionally designed to increase participant engagement and support educational objectives, details of which are provided in the section on gamified virtual reality for museums. Conversely, the non-gamified virtual reality version offers an informative and immersive experience centered on the knowledge of ancient fabrics, preservation techniques, and visual immersion, deliberately omitting the application of gamification concepts and gaming elements.

### 9.2. Measurement of Engagement, Motivation, and Knowledge Acquisition

To measure the effectiveness of the two types of approaches to virtual reality, we considered evaluating engagement, motivation, and knowledge acquisition. In terms of engagement, it was measured by how engaged visitors were with the virtual museum. The motivation aspect aimed to measure the effect of different simulated environments on users. Finally, knowledge acquisition was evaluated in terms of the learning outcomes that each approach could enhance. Details of each questionnaire are described below.

9.2.1. Questionnaire of Engagement

For gauging participant engagement, the user engagement scale (UES) [66] was implemented, as shown in Appendix A. The UES is a comprehensive instrument that measures

engagement across multiple dimensions: focused attention, perceived usability, aesthetic appeal, and the reward factor. Each statement on the UES is rated on a Likert scale from 1 (strongly disagree) to 5 (strongly agree), allowing participants to express the extent of their agreement with statements related to their engagement with the VR experience.

### 9.2.2. Questionnaire of Motivation

Participant motivation was measured using the intrinsic motivation inventory (IMI) [67], which assesses several key facets of motivation, including interest/enjoyment, perceived competence, effort/importance, and pressure/tension, as shown in Appendix B The IMI helps in understanding the intrinsic motivation of participants by asking them to rate statements on a Likert scale from 1 (strongly disagree) to 5 (strongly agree), reflecting their personal experience and motivational state during the activity.

### 9.2.3. Knowledge Acquisition

To evaluate knowledge acquisition, a set of 20 questions was developed collaboratively by lecturers from Chiang Mai University, museum archivists, and librarians. These questions were designed to measure participants' knowledge before and after the VR experience, serving as a pre-test and post-test. The questions were carefully crafted to cover key informational aspects of the museum's content, ensuring a robust assessment of the educational impact of the VR applications.

### *9.3. Procedure*

Our research procedure consisted of four steps designed to compare the non-gamified and gamified virtual reality groups. Each group completed the assessment using digital questionnaires on their own mobile devices. An outline and overview of the research procedure flow are presented in Figure 7. A detailed description of each step is provided below.

### Step 1: Recruitment and Pre-Testing

Recruitment for the study was conducted via social media channels associated with the Faculty of Humanities and the Department of Library and Information Science at Chiang Mai University. A total of 76 undergraduate students were successfully recruited. Each student received financial compensation of THB 100 (approximately USD 3) for their participation. Prior to the commencement of the study, participants completed a pre-test designed to evaluate their baseline knowledge of the museum's content. Consent forms were distributed and collected to ensure that all participants were thoroughly informed about the nature of the study and their role within it.

### Step 2: Group Assignment and Briefing

Participants were randomly assigned to one of two groups: the gamified virtual reality group or the non-gamified virtual reality group, each comprising 38 students. They were subsequently briefed on the research objectives, ensuring a comprehensive understanding of the study's goals and participant roles. In this phase, Meta 2 virtual reality devices (manufactured by Meta Platforms, Inc., Menlo Park, CA, USA) were introduced to the participants. Special attention was given to setting up the devices and making participants familiar with the equipment, including the adjustment of straps and the use of motion controllers. Participants were allotted approximately 20 min to become accustomed to the VR environment, which was crucial for minimizing disruptions during the testing phase and maximizing the quality of the user experience.

### Step 3: Conducting Tests in the Laboratory Setting

During this crucial phase of the study, the participants were provided with individual user logins to access the virtual reality application. To maintain the study's integrity, they were not previously informed about which version of the application, gamified or non-gamified, they would be using. The tests were conducted in a meticulously controlled

laboratory environment located within the Department of Library and Information Science. The laboratory was outfitted with three virtual reality devices to ensure that multiple participants could be tested concurrently. Each participant was granted a one-hour session to interact with the VR application, affording them sufficient time to fully engage with the content and functionalities specific to their assigned version. The entire testing stage was conducted over the course of one week, permitting a comprehensive and deliberate evaluation of each participant's experience with the virtual reality application.

Step 4: Post-Experience Questionnaires

After the VR experience, the participants were given questionnaires with which to measure various outcomes of the study, including their motivation, level of engagement, knowledge acquisition, and overall experience. The questionnaires were administered to gather both qualitative and quantitative data pertinent to the study's objectives. The data collected from the pre-tests and post-experience questionnaires were then analyzed to determine the impact of gamification elements on the educational and experiential aspects of the virtual museum visit. The analysis aimed to discern significant differences in the learning outcomes and engagement levels between the gamified and non-gamified VR groups.

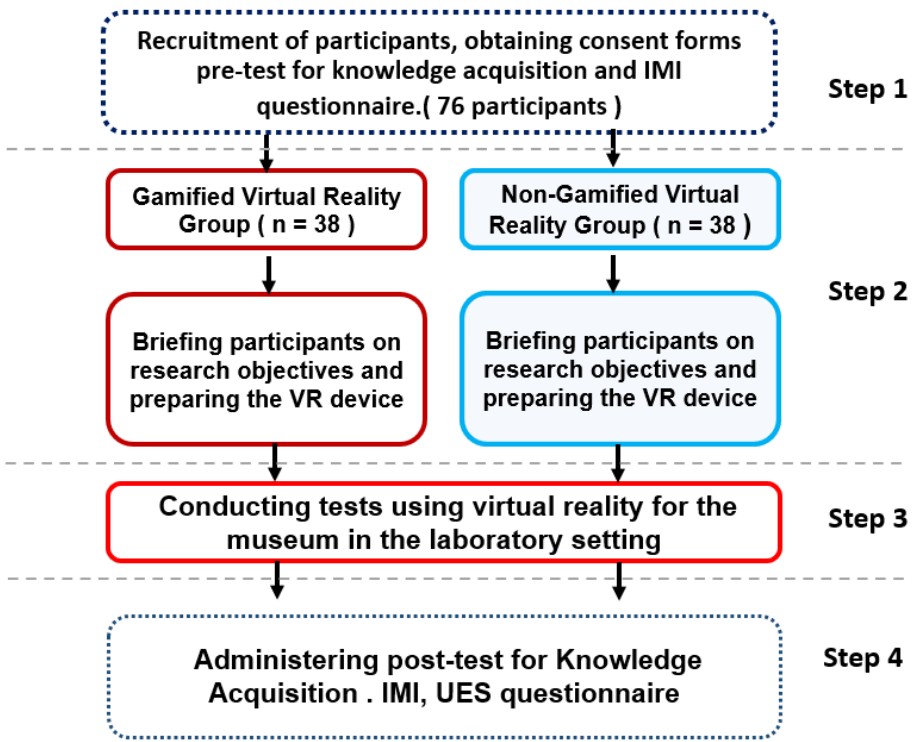

**Figure 7.** Overview of the research procedure.

## 10. Results and Data Analysis

### 10.1. Results of the Pre-/Post-Tests for Knowledge Acquisition

The pre- and post-test results for knowledge acquisition, as shown in Table 5 and Figure 8, revealed that both the non-gamified and gamified virtual reality groups experienced statistically significant improvements in knowledge, with $p$-values less than 0.001. The non-gamified VR (control) group, consisting of 38 participants, demonstrated a substantial increase from a mean pre-test score of 6.94 (SD = 2.30) to a post-test score of 9.10 (SD = 3.21). This indicates that even without the incorporation of gamified elements, the virtual reality experience contributed positively to knowledge acquisition. The gamified VR (experimental) group, also with 38 participants, showed a notable enhancement, with the mean score escalating from 6.18 (SD = 2.56) in the pre-test to 10.02 (SD = 3.46) in the

post-test. The significant improvements in both groups underscore the value of virtual reality as an educational tool, highlighting that the application of gamification elements, while beneficial, is not the sole factor in enhancing learning within virtual environments.

**Table 5.** Results of *t*-tests of knowledge acquisition of the pre- and post-tests for non-gamified and gamified virtual reality.

| Group | N | Pre-Test (SD) | Post-Test (SD) | *t*-Value | *p*-Value |
|---|---|---|---|---|---|
| Non-gamified VR (control) | 38 | 6.94 (2.30) | 9.10 (3.21) | 3.689 | *p* < 0.001 |
| Gamified VR (experiment) | 38 | 6.18 (2.56) | 10.02 (3.46) | 5.681 | *p* < 0.001 |

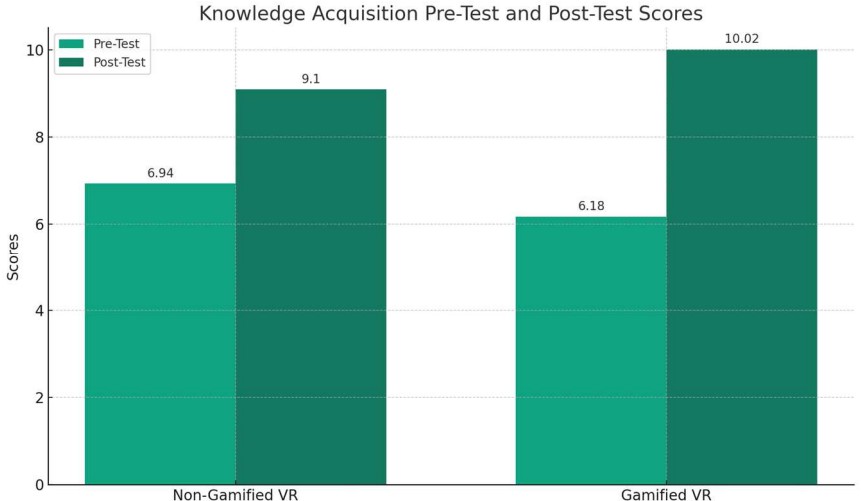

**Figure 8.** Result of knowledge acquisition pre-test and post-test.

### 10.2. Results of the IMI Questionnaires

The assessment of motivation, conducted using the intrinsic motivation inventory (IMI) and presented in Table 6 and Figure 9 for both the non-gamified and gamified virtual reality groups, revealed notable differences in motivational enhancement. The gamified VR group displayed a significantly greater improvement in the dimension of effort compared to the non-gamified VR group, with a notable increase in their mean scores (from 3.02 to 3.71, *t*-value = 3.77, *p* < 0.001). This suggests that the gamification elements may have particularly enhanced participants' willingness to exert effort during the VR experience. In contrast, both groups showed significant improvements in the dimension of interest. The non-gamified VR group's interest scores increased from a mean of 3.02 to 3.71, and the gamified VR group's scores similarly rose from 3.13 to 3.78. These substantial increases in both groups indicate that, irrespective of gamification, the virtual reality experience itself was a strong motivator in terms of arousing interest among the participants.

**Table 6.** Results of the IMI questionnaires and t-test of the non-gamified and gamified virtual reality groups.

| Group | Dimension | N | Pre (SD) | Post (SD) | Mean Difference (Post–Pre) | *t*-Value | *p*-Value |
|---|---|---|---|---|---|---|---|
| Non-gamified VR (control) | Perceived competence | 38 | 3.04 (0.73) | 3.21 (0.52) | 0.15 | 2.22 | 0.032 |
| | Interest | 38 | 3.02 (0.71) | 3.71 (0.61) | 0.68 | 4.66 | *p* < 0.001 |
| | Effort | 38 | 2.97 (0.71) | 3.10 (0.79) | 0.13 | 2.36 | 0.023 |
| Gamified VR (experiment) | Perceived competence | 38 | 3.10 (0.72) | 3.26 (0.60) | 0.15 | 2.63 | 0.012 |
| | Interest | 38 | 3.13 (0.81) | 3.78 (0.70) | 0.65 | 3.61 | *p* < 0.001 |
| | Effort | 38 | 3.02 (0.78) | 3.71 (0.65) | 0.68 | 3.77 | *p* < 0.001 |

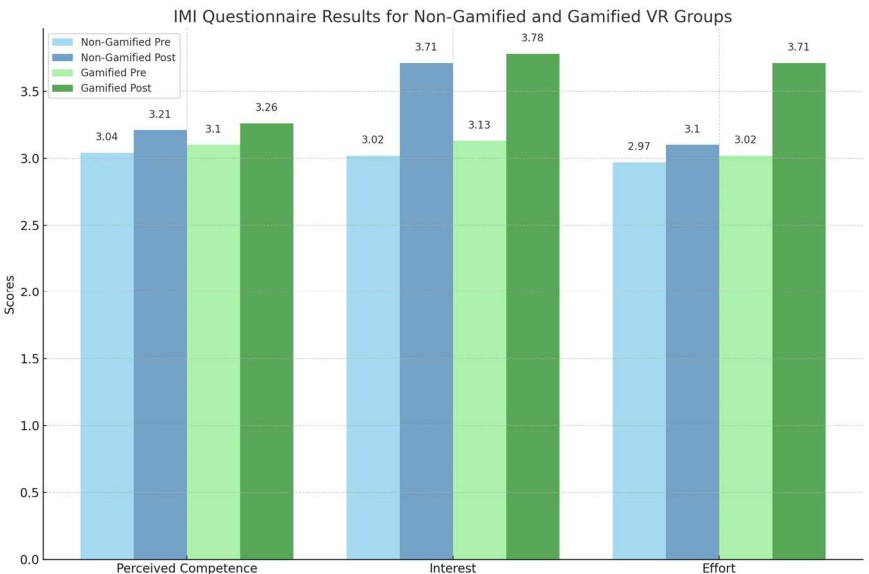

**Figure 9.** Result of the IMI questionnaire for the non-gamified and gamified VR groups.

### 10.3. Results of User Engagement

User engagement scale (UES) questionnaires were utilized to assess the user engagement levels in both the non-gamified and gamified virtual reality groups. The independent samples *t*-test results for these questionnaires are summarized in Table 7 and Figure 10. In our analysis, a *p*-value of less than 0.05 was considered statistically significant. In the dimension of focus attention, both the non-gamified VR (control) group and the gamified VR (experimental) group showed similar levels of engagement, with mean scores of 3.39 (SD = 0.88) and 3.42 (SD = 0.97), respectively, and a nonsignificant difference ($t = -0.123$, $p = 0.902$). Similarly, for perceived usability, there was no significant difference between the groups (non-gamified VR: mean = 2.97, SD = 0.78; gamified VR: mean = 2.84, SD = 0.85; $t = 0.698$, $p = 0.488$). Aesthetic appeal also did not show a significant difference, with the non-gamified group scoring a mean of 3.05 (SD = 0.76) and the gamified group scoring 3.39 (SD = 0.63) ($t = -0.299$, $p = 0.766$).

However, in the dimension of reward, the gamified VR group exhibited significantly higher engagement compared to the non-gamified VR group. The gamified group had a mean score of 3.57 (SD = 0.94), while the non-gamified group scored 3.05 (SD = 0.80), resulting in a statistically significant difference ($t = -2.610$, $p = 0.011$). This indicates that the integration of gamification elements significantly enhanced the perception of reward among participants in the gamified VR group.

**Table 7.** Results of UES questionnaires using independent samples *t*-tests.

| UES Questionnaires | Group | N | Mean Score (SD) | *t* | *p*-Value |
|---|---|---|---|---|---|
| Focus attention | Non-gamified VR (Control) | 38 | 3.39 (0.88) | −0.123 | 0.902 |
| | Gamified VR (experiment) | 38 | 3.42 (0.97) | | |
| Perceived usability | Non-gamified VR (control) | 38 | 2.97 (0.78) | 0.698 | 0.488 |
| | Gamified VR (experiment) | 38 | 2.84 (0.85) | | |
| Aesthetic appeal | Non-gamified VR (control) | 38 | 3.05 (0.76) | −0.299 | 0.766 |
| | Gamified VR (experiment) | 38 | 3.39 (0.63) | | |
| Reward | Non-gamified VR (control) | 38 | 3.05 (0.80) | −2.610 | 0.011 * |
| | Gamified VR (experiment) | 38 | 3.57 (0.94) | | |

Note: * *p*-value less than 0.05.

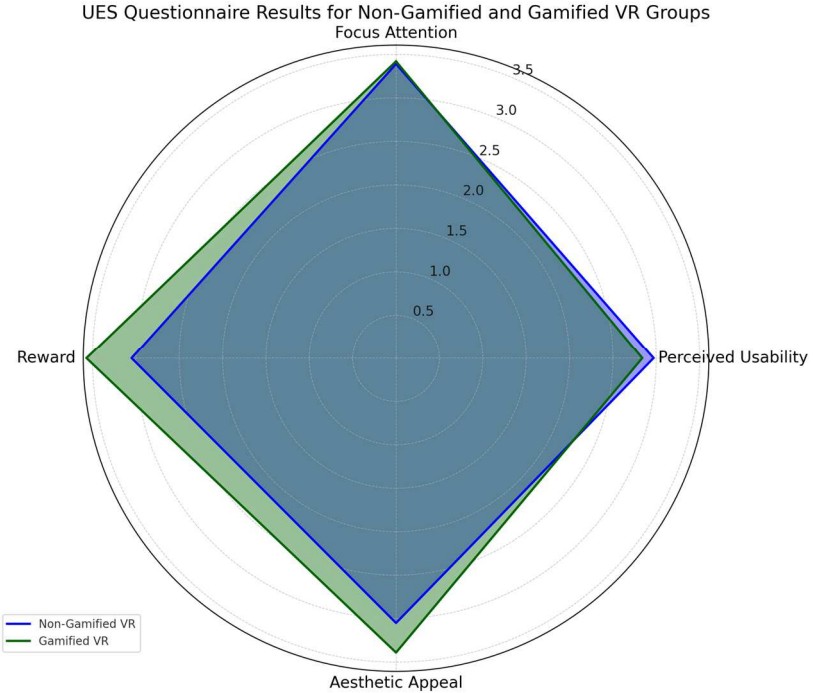

**Figure 10.** Results of UES questionnaires for the non-gamified and gamified VR groups.

## 11. Discussion and Findings

**RQ1.** *Effect of Gamified VR Experience on Motivation*

The analysis of the intrinsic motivation inventory (IMI) results, as detailed in Table 6 and Figure 9, revealed significant improvements in both the non-gamified and gamified VR groups, with a notable increase in the interest dimension. In particular, the gamified VR group showed an increased effort level. This suggests that gamification elements in VR, like achievements and interactive challenges, effectively enhance participants' willingness to engage and invest effort in the virtual museum experience. These experiences, designed based on the digital twin concept of the Wieng Yong House Museum in Lamphun, Thailand, align with previous research that has applied gamification in museum contexts to boost visitor motivation [61,68]. Furthermore, studies [69] incorporating game elements such as challenges, feedback, rewards, and leaderboards have also been shown to improve visitor satisfaction. Our findings highlight that gamified virtual reality can significantly enhance intrinsic motivation, particularly in the effort dimension, encouraging visitors to engage in activities for their inherent satisfaction rather than for external rewards. This has significant implications for educational endeavors, highlighting the potential of gamified VR to enrich learning experiences.

**RQ2.** *Effect of Gamified VR Experience on Engagement*

The user engagement scale (UES) results provided nuanced insights into user engagement. Both the non-gamified and gamified VR experiences scored similarly in the dimensions of focused attention, perceived usability, and aesthetic appeal, as detailed in Table 7 and Figure 10; however, the gamified VR group demonstrated significantly higher scores in the reward dimension. This finding emphasizes the effectiveness of gamification elements in enhancing the perception of reward, thus making the experience more engaging and satisfying for users. These results are consistent with previous research that has applied gamification in museum contexts to improve visitor engagement [70–72], suggesting that incorporating these elements can substantially elevate the overall visitor experience.

**RQ3.** *Learning Outcomes from Implementing Gamification in VR*

The analysis of pre-test and post-test results for knowledge acquisition showed significant improvements in both the non-gamified and gamified VR groups, as detailed in Table 5 and Figure 8; however, the gamified VR group did not show a significantly greater improvement in knowledge acquisition than the non-gamified VR group, contrary to expectations given the higher post-test scores observed. While direct comparisons in museum settings are scarce, related research in educational and training contexts suggests that gamification can improve learning outcomes compared to virtual reality alone [73–75]. Additionally, many studies have reported better knowledge retention with gamification [76–78], which could indicate the potential benefits of applying gamification in virtual reality within museum contexts. Nevertheless, our findings suggest that implementing gamification concepts in virtual reality may not always yield more effective learning outcomes. This underscores the need for a nuanced approach to integrating gamification into VR for educational purposes, taking into account the specific context and learning objectives.

## 12. Conclusions

This study aimed to develop the digital twin concept for the Wieng Yong House Museum in Lamphun, Thailand, by digitizing heritage objects and exploring the effects of implementing gamification in virtual reality (VR) within museum settings. To this end, we compared gamified and non-gamified VR groups, assessing their impact on participant engagement, motivation, and learning performance. Two distinct versions of VR applications were developed: a gamified version, incorporating interactive game elements such as leaderboards, progression, badges, profiles, achievements, and quizzes, and a non-gamified version, devoid of these elements.

Our findings revealed that while gamification significantly enhances the reward dimension of visitor engagement, its impact on motivation is predominantly observed in the effort dimension, and its effect on learning outcomes may not be as profound. This study contributes to the growing body of knowledge on how gamification can be applied in virtual reality applications within museums. It highlights the nuanced impact of gamified elements on the user experience and underscores the importance of a balanced approach in integrating these elements. Such integration should be carefully considered, keeping in mind the specific objectives and contextual needs of the museum setting.

In conclusion, this study synthesizes findings and formulates general guidelines for researchers, librarians, and practitioners considering the implementation of gamification in virtual reality within museum settings. These guidelines are based on insights gained from our research at the Wieng Yong House Museum in Lamphun, Thailand.

- Integrating gamification concepts in virtual reality for museum contexts may not significantly enhance overall motivation and engagement but could improve specific dimensions such as effort in motivation and reward in engagement. The impact of these single dimensions might be sufficient justification for implementing gamification concepts to enhance visitor experiences, particularly when considering the costs involved.
- In terms of learning outcomes, both non-gamified and gamified VR groups demonstrated significant improvements in knowledge acquisition, with no marked difference between the two. If the primary goal is educational, implementing gamification may not be necessary; however, future research could explore the effects of gamification on knowledge retention in virtual reality applications in museum settings.
- Based on our development experience of the digital twin concept for the Wieng Yong House Museum and observations of participant interactions in both the non-gamified and gamified VR groups, we recommend considering the implementation of gamification. The benefits in terms of motivation and engagement could outweigh the relatively low effort and cost of incorporation. Additionally, we noted that participants in the gamified group tended to spend more time and exhibit greater satisfaction with the virtual museum experience. This suggests that, even with minimal implementation

efforts, gamification can positively influence user interaction and enjoyment in virtual museum environments.

## 13. Future Research and Limitations

This study, while providing valuable insights into the implementation of gamification in virtual reality within museum settings, has several limitations. Firstly, the participant sample was drawn exclusively from undergraduate students at a single university, which may limit the generalizability of the findings. Secondly, the specific cultural and historical context of the Wieng Yong House Museum in Lamphun, Thailand, means that the results may not be directly applicable to other museums with different themes or visitor demographics. Additionally, the study focused solely on short-term engagement and learning outcomes without assessing long-term retention or the potential for repeated visits.

Future work, reflecting on our findings and acknowledging the limitations of our participant sample, as well as the focus on short-term outcomes, should aim for broader diversity in participants to enhance the applicability of the findings. There is a need to explore the long-term impacts of gamification on engagement and knowledge retention and to extend research across various cultural and thematic museum contexts. Integrating VR with emerging technologies, such as AR, MR, and AI, could further innovate visitor experiences. Conducting an economic analysis to evaluate the costs and benefits of gamified VR experiences in museums, alongside qualitative methods to capture visitors' subjective experiences, could provide deeper insights. These efforts will refine our understanding of gamification's role in enriching museum experiences and guiding more effective, engaging, and accessible cultural heritage interactions.

**Author Contributions:** Conceptualization, S.C. and K.P.; methodology, K.P. and S.Y.; software, K.I.; validation, K.P., S.C. and S.Y.; formal analysis, K.P.; investigation, K.P.; resources, K.P. and S.Y.; data curation, N.W.; writing—original draft preparation, K.P. and N.W.; writing—review and editing, S.C. and K.P.; visualization, K.P.; supervision, P.A.; project administration, K.I.; funding acquisition, K.P. All authors have read and agreed to the published version of the manuscript.

**Funding:** This research project is partially supported by Chiang Mai University and the Faculty of Humanities, Chiang Mai University.

**Data Availability Statement:** Data are contained within the article.

**Conflicts of Interest:** The authors declare no conflicts of interest.

## Appendix A

**Table A1.** The user engagement scale questionnaire (five-point Likert scale ranging from strongly disagree to strongly agree).

| Dimension | Questionnaire |
| --- | --- |
| Focused attention | The time I spent using virtual reality technology just slipped away. |
| | I was absorbed in this experience. |
| | I felt frustrated while using this virtual reality technology. |
| Perceived usability | I found this virtual reality technology confusing to use. |
| | Using this virtual reality technology was taxing. |
| | This virtual reality technology was attractive. |
| Aesthetic appeal | This virtual reality technology was aesthetically appealing. |
| | This virtual reality technology appealed to my senses. |
| Reward | Using virtual reality technology was worthwhile. |
| | My experience was rewarding |
| | I felt interested in these experiences |

## Appendix B

**Table A2.** Intrinsic motivation inventory questionnaire (five-point Likert scale ranging from strongly disagree to strongly agree).

| Dimension | Questionnaire |
|---|---|
| Perceived competence | I think I was good at learning through virtual reality. |
| | I think I did pretty well in learning through virtual reality. |
| | I am satisfied with my performance while learning through the virtual reality. |
| | I was pretty skilled at learning through virtual reality. |
| | I think I was pretty good at learning through virtual reality. |
| Interest | I think learning through virtual reality was quite Enjoyable. |
| | I think learning through virtual reality was interesting. |
| | I think learning through virtual reality was fun. |
| | While I was learning through the virtual reality, I often thought about how much I enjoyed it. |
| | I think learning through mixed reality was boring. |
| Effort | I did my best while I was learning through the mixed reality. |
| | I tried very hard to do well in learning through mixed reality. |
| | It was important to me to do well in learning through mixed reality. |
| | I put a lot of effort into making this mixed reality. |

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
