# Peer review of "Comparing the Impact of Non-Gamified and Gamified Virtual Reality in Digital Twin Virtual Museum Environments: A Case Study of Wieng Yong House Museum, Thailand"

_heritage, doi:10.3390/heritage7040089_

Round 1

Reviewer 1 Report

Comments and Suggestions for Authors

The text is well written and the methodology is correctly presented. The state of the art demonstrates knowledge of the subject and the citations are appropriate. The limitations of the work are correctly described and the results, which slightly contradict the initial hypothesis, are presented clearly. Maybe the title doesn't reflect the results, but in the end, the work was well done.

Author Response

Thank you very much for your suggestion. Based on your feedback, we have decided to revise the title to “Comparing the Impact of Non-Gamified and Gamified Virtual Reality in Digital Twin Virtual Museum Environments: A Case Study of the Wieng Yong House Museum, Thailand.”.

Reviewer 2 Report

Comments and Suggestions for Authors

This paper examines the effects of gamified and non-gamified VR experiences within virtual museum environments, highlighting the concept of a digital twin and its focus on cultural heritage. Even though it does not present groundbreaking research, this paper deserves some merit, and I suggest publishing it after a major revision procedure. In the introduction section, the authors should better frame the meaning of digital twins. I would recommend considering the following papers: https://doi.org/10.3390/infrastructures8050086 https://doi.org/10.1016/j.engstruct.2022.115256. At the end of the introduction section, I suggest introducing the paper's outline to improve its readability. English should be polished. I would suggest representing the project goal with figures summarising the integrated workflow. A bullet points conclusions section is recommended. I also want to offer to introduce a future work paragraph.

Some issues with references.

Comments on the Quality of English Language

English should be polished.

Author Response

Thank you for your suggestions. We have expanded the introduction to provide a definition of digital twins by integrating insights from the suggested articles and added a detailed outline of the paper at the end of the introduction section.

Reviewer 3 Report

Comments and Suggestions for Authors

In its current form, the article gives the impression that it is unfinished. The References section is found twice with some differences. The numbering of bibliographic references in the text is different. Also, some sentences must be revised because they are either incomplete or common truisms. The text requires careful review by the authors in order to enhance the coherence and the flow of reading. There are a substantial number of incorrect words separated by hyphens. There article should focus more on the methods of measuring the impact of the two type of approaches. The concept of digital twin is relation to cultural heritage goods is not explained and doesn’t seem to receive any interest from the authors. In fact the paper doesn’t refer to a review of the general comparison of the impact generated by Non-Gamified and Gamified Virtual Reality Museum Environments - it presents a single case study.

 The article can be highly improved. The subject is interesting but it must be understood from different perspectives: education, cultural heritage science, technological development, sociocultural context. The case study is very narrow and it doesn’t reflect the title which is very pretentious.

27-29 The first phrase which opens the article includes an obvious repetition. 

29-30 After "Notably, cultural heritage organizations are investing in VR" please name some examples which are most relevant.

33 What does "unnusual narratives" mean in this case? Please give some examples. How are these type of narratives enhance the value of a museum. Try to develop this idea a little bit more.

38-39 "Moreover, as VR technology becomes more affordable and accessible for personal use, it creates new opportunities for active participation in cultural heritage" I consider that active participation in cultural heritage is strongly related to the aspects related to the physical presence of the cultural goods. In this case is a virtual participation. Please rephrase.

41-42 "offering a unique approach to education and engagement that bridges the virtual and physical realms." Please give some examples.

45-46 "To illustrate, [9] indicated that gamification had positive effects on cognitive, affective, and behavioral learning outcomes." The phrase doesn’t make sense and the reference the looks like is put by default.

59-61 This phrase is repeating without bringing new information. Try to explain and to give some examples of museum narratives. It would help the reader understand better 

68-69 "Moreover, scholars have demonstrated that well-developed VR content not only motivates students but also allows for in-depth exploration of a topic [23]". Try to develop/explain more on how VR motivates students and allows for in-depth exploration of a topic. Use multiple references. Please mention which are the categories of users that use VR and why

In the Section Literature Review, especially in 2.2. Gamification  see also the following works:

C. Öztürk & Ö. Korkmaz, The Effect of Gamification Activities on Students' Academic Achievements in Social Studies Course, Attitudes towards the Course and Cooperative Learning Skills, Participatory Educational Research (PER), Vol. 7(1), pp. 1-15, March 2020 http://dx.doi.org/10.17275/per.20.1.7.1 

M. C. Lister, Gamification: The effect on student motivation and performance at the post-secondary level”, Issues and Trends in Learning Technologies., 3(2), https://doi.org/10.2458/azu_itet_v3i2_lister 

T. A. Papp, Gamification Effects on Motivation and Learning: Application to Primary and College Students, International Journal for Cross-Disciplinary Subjects in Education (IJCDSE), Volume 8, Issue 3, September 2017 https://infonomics-society.org/wp-content/uploads/ijcdse/published-papers/volume-8-2017/Gamification-Effects-on-Motivation-and-Learning.pdf 

N. A. Boudadi & M. Gutiérrez-Colón, Effect of Gamification on students’ motivation and learning achievement in Second Language Acquisition within higher education: a literature review 2011-2019The EUROCALL Review, Volume 28, No. 1, March 2020, https://files.eric.ed.gov/fulltext/EJ1257523.pdf 

O’Connor, S.; Colreavy-Donnelly, S.; Dunwell, I. Fostering Engagement with Cultural Heritage Through Immersive VR and Gamification; Springer Series Cultural Computing; Springer International Publishing: Cham, Switzerland, 2020; pp. 301–321.

Dicheva, D.; Dichev, C.; Agre, G.; Angelova, G. Gamification in education: A systematic mapping study. J. Educ. Technol. Soc. 2015, 18, 75–88.

N. Z. Legaki, N. Xi, J. Hamari, K. Karpouzis, V. Assimakopoulos, The effect of challenge-based gamification on learning: An experiment in the context of statistics education, International Journal of Human-Computer Studies, Volume 144, 2020, 102496, https://doi.org/10.1016/j.ijhcs.2020.102496. 

Gamification is a method used in pedagogy and is used in all the fields related to education. Education in the field of Cultural heritage is just one of the domains. The main goal is to capture the attention and to be able to transmit information. Please see the reference list provided 

For Table 1. Please provide a better explanation regarding the way you determined the categories Learning effects and Player experience

141-142 "Learning of cultural heritage". What does it mean? To recognize it? To be aware of this particular category of artefacts

142-144 What about the quality of the information that is provided about the cultural heritage concept?

Starting from: 4. Virtual reality design and Implementation – until the end the authors are presenting a single study although, the title refers to a general comparison of the impact between Non-Gamified and Gamified Virtual Reality Museum Environments. I think that the title should reflect that this research is based on a single case-study and name it. It gives the impression that is a review 

Are there any statistics regarding visitors of this particular Wieng Yong House Museum?  Can the results of this study enhance the visibility of the museum and how?

6. Reconstruction and Digitalization of Heritage Objects in Museums – How is the impact affected by these characteristics – quality of 3D models (digital models). Can you quantify them?

8. Research methodology Are the participants of this study relevant for the purpose of the research? Can you establish the impact by consulting a uniform category of participants in relation to the museums heritage? 

11. Discussion and finings - findings

The data from the Tables can be further used in diagram to highlight the results of this study?

Comments on the Quality of English Language

Please read the text carefully. There are many spelling mistakes. In some parts it seems that the text was never finished. Pay special attention to punctuation. It looks like the text was formatted in another program

Author Response

We have attached the document containing a point-by-point response to the reviewers' comments and concerns."

Reviewer 4 Report

Comments and Suggestions for Authors

The article is very clear and well structured, even if in some places it is a little repetitive and is lacking in descriptive detail with regard to the realisation of the Non-Gamified Virtual reality version.

There are a few typographical corrections to be made, in particular in the introduction, line 29 repeats the same concept as above, line 81 is missing the expansion of the acronym AR and also in table no. 1 for the acronym HDMs, which is only later explained. It is also unclear why there is a double bibliography, I guess that from line 520 to line 666 should be removed.

Author Response

Thank you for pointing this out. We have revised the introduction to eliminate redundancy and have added the full expansions for the acronyms "AR" as "Augmented Reality" and "HMDs" as "Head-Mounted Displays" 

Round 2

Reviewer 2 Report

Comments and Suggestions for Authors

Accept in the present form

Comments on the Quality of English Language

Accept in the present form

Reviewer 3 Report

Comments and Suggestions for Authors

I consider that the quality of the paper has increased since revision and I find the article to be relevant  for the topic. I am glad that the authors took into account the recommendations. I hope it was useful to them.